# Mathematical AI-Driven Insights into Societal Dynamics and Resilience

Md Sarwar Kamal
mdkamal@csu.edu.au
Charles Sturt University
Albury, NSW, Australia

Md Rafiqul Islam
mislam@csu.edu.au
Charles Sturt University
Albury, NSW, Australia

## Abstract

The behavioral patterns of individuals within a society can serve as a reflection of its overall conditions and status. By examining these patterns, we can identify critical societal issues such as poverty, hunger, crime, sadness, underdevelopment, premature mortality, and social structure. Our research aims to understand the factors associated with these problems by leveraging societal data analytics. We explore societal behavior and resilience by analyzing Hindi cinema from 1951 onwards and comparing it with contemporary community data from developed countries. This is achieved through integrated interpretive and mathematical artificial intelligence (MAI) techniques, alongside natural language processing (NLP). The primary goal is to uncover and analyze the underlying societal norms, resilience patterns, and behavioral dynamics depicted in Hindi movies, and to validate these findings against real-world community data. The methodology begins with language processing techniques, such as term frequency analysis and contextual thematic analysis, to extract and quantify thematic variables from song lyrics. These variables capture fundamental societal issues like wealth disparity, human suffering, and societal despair. Additionally, Fourier and Laplace transforms are used for time-series analysis of audio signals and thematic continuity in video sequences. Game theory models are also applied to study decision-making processes and social interactions portrayed in the films. To ensure the transparency and interpretability of the MAI-driven insights, we employ explainable AI (XAI) approaches, including counterfactual explanations, feature visualization, and concept activation vectors (CAVs).From both the outcomes of MAI and XAI, we have noticed that there are some key features showing from both the data set and these are hunger, crimes, underdevelopment, poverty, and social structure. By treating multimedia content as a reflection of societal views and comparing it with empirical data, this research provides a core analysis of societal dynamics across different cultural contexts. Our proposed methods demonstrate higher accuracy compared to state-of-the-art approaches which ensure our findings more explainable and insightful.

## CCS Concepts

• **Mathematical AI → Explainable AI**.

## Keywords

Societal behavior, resilience, multimedia content analysis, explainable AI, interpretable AI, Hindi cinema, community data, cultural studies, public policy

**ACM Reference Format:**
Md Sarwar Kamal and Md Rafiqul Islam. 2025. Mathematical AI-Driven Insights into Societal Dynamics and Resilience. In *Proceedings of The Web Conference 2025 (WWW)*. ACM, New York, NY, USA, 8 pages. https://doi.org/10.1145/3701716.3717737

## 1 Introduction

Understanding social behaviors and resilience through multimedia content and community data presents a unique opportunity to bridge the gap between theoretical social science research and practical, data-driven insights [1, 2, 3]. The cultural richness of Hindi films, spanning more than seven decades, serves as a profound mirror of the social norms, values, behavioral dynamics and resilience patterns inherent in Indian society. These films not only reflect the evolution of societal attitudes and responses to various challenges, but also provide a fertile ground for analyzing the changes in cultural narratives over time. Conversely, the other developed community data provides a snapshot of contemporary societal behaviors and resilience and represents an entirely different cultural context characterized by diverse demographics and societal structures [4, 5, 6, 7]. This dataset allows researchers to assess current social health, community resilience and behavioral trends across a developed, multicultural society. The juxtaposition of historical and cultural insights from the Hindi film with empirical data from Europe allows for a comparative analysis that can illuminate both universal and culturally specific patterns of social behavior and social resilience.

Despite the vast potential of these sources, existing research often falls short in effectively link diverse multimedia content with empirical community data in order to investigate complex social phenomena [8, 9, 10]. Social science research methods are usually based on qualitative analysis or conventional statistical methods, which may not suffice to unravel the complex patterns inherent in rich multimedia content or large community datasets. In addition, there is a high reliance on data analysis methods that lack transparency and interpretability, which can confuse non-experts and hinder the practical application of research findings.

## 1.1 Research Questions

This research is guided by several pivotal questions that seek to deepen our understanding of societal dynamics through innovative analytical lenses:

- How can interpretative and MAI methods be used effectively to analyze social behaviors and resilience portrayed in multimedia content?
- What insights into social norms and patterns of resilience can be gained by comparing cultural representations in Hindi films with empirical data from other developed communities?
- How can these insights inform public policy, contribute to socio-cultural studies and strengthen the resilience of communities?

## 1.2 Contributions

The contributions of this research are diverse and aim to reshape the landscape of sociocultural analytics:

- We pioneer the integration of advanced interpretative AI techniques such as Fourier and Laplace transforms for time series analysis of audio signals and thematic continuity in video sequences, along with game-theoretic models for analyzing decision-making processes and social interactions depicted in Hindi films.
- This research introduces a novel comparative framework that employs state-of-the-art mathematical AI methods, including counterfactual explanations, feature visualization, and CAVs, to improve the transparency and interpretability of AI-based insights.
- Our research findings are designed to be easily accessible so that not only academics, but also policy makers and practitioners can understand and effectively implement the results.
- The research provides new perspectives on the impact of cultural artifacts on societal behaviors and resilience and provides a comprehensive model for future research in global sociocultural studies.

## 2 Literature Review

Research exploring the intersection of AI and societal dynamics has gained significant momentum in recent years, particularly with the integration of mathematical AI techniques to analyze complex social phenomena. Theoretical foundations for societal behavior modeling have traditionally relied on qualitative approaches, but advancements in computational methods have enabled more quantitative and interpretable analyses. Carl et al. [11] reflect on how the movement began, who was involved, how momentum was built and sustained, our culture and organisational methods, the challenges we've faced, and what we have learnt in the process. To bring this to life, examples from the work of psychologists for social change btween two of our autonomous geographical groups are explored.In COVID-19 pandemic has caused a rise in stress, mental health concerns, and externalizing behaviors in society and their caregivers across the globe and illuminated the need to reduce stress levels and support self-regulation skills in even the youngest of children. Bockmann et al. [12] describe the use of mindfulness-based interventions (MBIs) to support young children's self-regulation in early childhood settings. A total of 18 research studies conducted between 2010 and 2021 were identified. The main purposes of the studies reviewed were to examine the effects of MBIs on the development of emotional, behavioral, and cognitive self-regulation. Ariyani et al. [13] investigates the effects of online counseling platforms and virtual community support on work-related stress management among young professionals in Jakarta. A quantitative methodology was employed, gathering data from 150 participants via a Likert scale (1-5), and analyzed with SPSS version 26.

## 3 Data Collection

We collected data from publicly available Hindi films and publicly available community data using a robust data collection methodology that includes multimedia content and community-level datasets to provide a comprehensive insight into societal behaviors and resilience patterns. Our implementation and datasets can be accessed on GitHub(https://github.com/Sarwar899030/Societal-Dynamics-and-Resilience).

**Multimedia Content from Hindi Cinema**

We have focused specifically on Hindi cinema as a reflection of social norms and values, concentrating on songs from films released since 1951. Songs in Hindi films often serve as cultural commentaries that reflect the social, political and emotional zeitgeist of their times. For this study, we have carefully selected 100 songs from various films spanning seven decades. Each song was selected based on its thematic relevance to social issues and its resilience to ensure a balanced representation across different eras. Both audio and video components of these songs were analyzed to provide a complete picture of the messages conveyed. The list of films and songs with their respective years of release and thematic classifications can be found in the given link.

**Community Data from Developed Countries**

Community data includes publicly available datasets that reflect current societal behaviors and challenges in developed countries. These datasets are drawn from public records and focus on issues such as unemployment, accessibility of health services, crime rates, and community response to natural disasters. This data provides a real-time snapshot of the issues, challenges and resilience in these communities. Despite advances in technology and social policy, our preliminary analysis shows that many of the societal challenges documented in 1951 are still relevant today. This persistence of problems allows us to draw parallels and contrasts between the historical cinematic representations and the contemporary real-world data.

**Rationale for Data Choices**

The choice of Hindi songs from 1951 onwards and community data from developed countries was driven by the objective to explore and compare societal reflections across two distinct cultural settings and time periods. This comparative analysis helps in understanding not only the specific cultural responses to societal challenges but also in identifying universal themes and resilience mechanisms that transcend geographical and temporal boundaries. By integrating these diverse datasets, our research aims to provide a nuanced understanding of how societies express and tackle challenges through cultural artifacts like cinema, and how these expressions align with or differ from real-world data on societal resilience and challenges.

## 4 Methodology

Our proposed approach is illustrated in Figure 1. It starts with data collection from three types of public datasets: resilience data, song lyrics data, and audiovisual data. The resilience data is analyzed using black-box classifiers, which are opaque machine learning models that do not reveal their internal logic. The results are then processed using explainable artificial intelligence (XAI) approaches, in particular Minimum Redundancy Maximum Relevance (MRMR) and SHapley Additive exPlanations (SHAP), to ensure the transparency and interpretability of the insights gained. Song lyrics data will be subjected to term frequency analysis to quantify the prevalence of key thematic elements such as wealth disparity and human misery. Audio/video data is subjected to Fourier and Laplace transforms to analyze patterns in time series data of audio signals and video sequences. These are also analyzed using black box classifiers. The results of the classifiers used for lyrics and audio/video data are further analyzed using XAI approaches, including counterfactual explanations and concept activation vectors (CAVs). These approaches help to understand the impact of the different features on the model's predictions. The final results, presented as XAI results, offer visual and statistical interpretations that provide a deeper understanding of the social dynamics as depicted in the Hindi film compared to real community data. This comprehensive analysis provides a robust framework for examining social norms and resilience in different cultural settings.

### 4.1 Lyric Analysis for Societal Themes

To understand the social commentary embedded in the songs of the Hindi film, we apply text analysis techniques to selected song lyrics that address important social issues and challenges. The song lyrics selected for analysis vividly describe social inequalities, human conflicts and existential reflections that point to broader societal problems. We focus on identifying and quantifying specific themes such as wealth inequality, human misery and societal despair.

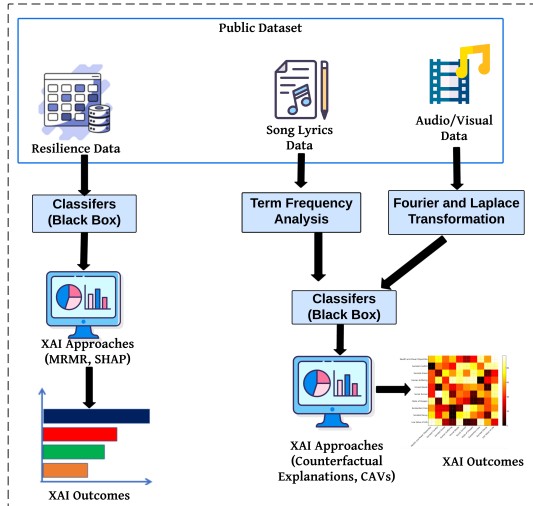

**Figure 1: A schematic diagram of the proposed framework for identifying societal dynamics and resilience.**

### 4.2 Contextual Thematic Analysis

We employ contextual thematic analysis to systematically identify and categorize lyrical content into thematic variables. This method combines qualitative content analysis with computational text analysis techniques.

*4.2.1 Variable Extraction from Lyrics.* We begin by extracting specific variables from the lyrics that symbolize various societal themes. This process involves both manual annotation and automated NLP techniques to ensure comprehensive coverage and accuracy. The thematic variables identified and their societal implications include:

- $V_{palaces}$ - Symbolizing wealth and power disparities, often depicted through references to opulent imagery such as palaces, thrones, and crowns.
- $V_{enemy\_of\_man}$ - Indicating societal structures in conflict with individual well-being, often reflected through narratives of societal opposition or systemic barriers.
- $V_{wealth\_hungry}$ - Reflecting societal greed, captured through critiques of materialism and the pursuit of wealth at the expense of ethical values.
- $V_{wounds}$ - Showcasing human suffering, either physical or emotional, highlighting the personal toll of broader societal issues.
- $V_{thirst}$ - Representing unmet needs or desires, symbolizing the existential yearning for meaning, justice, or essentials like water and love.
- $V_{confusion}$ - Denoting social or personal turmoil, often expressed through lyrics depicting uncertainty, doubt, or existential questions.
- $V_{sadness}$ - Indicating a state of despair, usually in response to personal loss or societal decay, often conveyed through melancholic tones.
- $V_{misery\_world}$ - Suggesting existential crisis or disillusionment, questioning the value or purpose of the world as it stands.
- $V_{dead\_colony}$ - Highlighting societal decay, portrayed through imagery of death and stagnation, suggesting a lack of progress or vitality.
- $V_{cheap\_death}$ - Emphasizing the low value placed on life within society, often through narratives where life is seen as less valuable than material or ideological gains.

*4.2.2 Quantitative Analysis.* To quantify the prevalence of each variable, Term Frequency (TF) analysis is used:

$$Q_{theme} = \sum_{i=1}^{n} \text{TF}(V_i) \tag{1}$$

where $Q_{theme}$ represents the quantified theme score for each song, providing a measure of how often each thematic element appears, indicating its significance within the lyrics.

*4.2.3 Thematic Continuity Analysis.* To assess thematic continuity within the audio and video sequences of the songs, we apply Laplace transforms:

$$L(s) = \int_0^\infty e^{-st} \cdot Q_{theme}(t)\, dt \tag{2}$$

This analysis helps in understanding how themes are introduced, developed, and concluded within cinematic contexts, reflecting temporal dynamics and evolution.

*4.2.4 Determine the Frequency Components.* The Fourier Transform is applied to determine the frequency components associated with each thematic variable, identifying how prevalently and in what contexts these themes appear in the audio:

$$F_{V_{wealth\_hungry}}(\omega) = \int_{-\infty}^{\infty} V_{wealth\_hungry}(t)e^{-j\omega t}\, dt \qquad (3)$$

$$F_{V_{wounds}}(\omega) = \int_{-\infty}^{\infty} V_{wounds}(t)e^{-j\omega t}\, dt \qquad (4)$$

$$F_{V_{thirst}}(\omega) = \int_{-\infty}^{\infty} V_{thirst}(t)e^{-j\omega t}\, dt \qquad (5)$$

$$F_{V_{confusion}}(\omega) = \int_{-\infty}^{\infty} V_{confusion}(t)e^{-j\omega t}\, dt \qquad (6)$$

$$F_{V_{sadness}}(\omega) = \int_{-\infty}^{\infty} V_{sadness}(t)e^{-j\omega t}\, dt \qquad (7)$$

$$F_{V_{misery\_world}}(\omega) = \int_{-\infty}^{\infty} V_{misery\_world}(t)e^{-j\omega t}\, dt \qquad (8)$$

These equations allow us to quantify the prevalence and variability of each theme over time in the songs. By analyzing the frequency components, we can infer how these thematic elements fluctuate, peak, or diminish throughout the audio tracks, offering insights into their emotional and narrative significance within the context of the songs.

*4.2.5 Specific thematic Expressions Identification.* Further, the Laplace Transform examines the evolution and decay of thematic expressions throughout the songs to study the development and resolution of specific thematic expressions throughout the song:

$$L_{V_{wealth\_hungry}}(s) = \int_{0}^{\infty} V_{wealth\_hungry}(t)e^{-st}\, dt \qquad (9)$$

$$L_{V_{wounds}}(s) = \int_{0}^{\infty} V_{wounds}(t)e^{-st}\, dt \qquad (10)$$

$$L_{V_{thirst}}(s) = \int_{0}^{\infty} V_{thirst}(t)e^{-st}\, dt \qquad (11)$$

$$L_{V_{confusion}}(s) = \int_{0}^{\infty} V_{confusion}(t)e^{-st}\, dt \qquad (12)$$

$$L_{V_{sadness}}(s) = \int_{0}^{\infty} V_{sadness}(t)e^{-st}\, dt \qquad (13)$$

$$L_{V_{misery\_world}}(s) = \int_{0}^{\infty} V_{misery\_world}(t)e^{-st}\, dt \qquad (14)$$

These equations enable us to understand the temporal dynamics and evolution of these themes within the cinematic context.

## 4.3 Game Theory Analysis

Game theory is used to model the interactions influenced by thematic variables, exploring how societal roles and expectations shape individual decisions:

$$\begin{aligned}\Gamma = \{\mathcal{N}, \\ \{S_{V_{wealth\_hungry}}, S_{V_{wounds}}, \ldots, S_{V_{misery\_world}}\}, \\ \{U_{V_{wealth\_hungry}}, U_{V_{wounds}}, \ldots, U_{V_{misery\_world}}\}\} \quad (15)\end{aligned}$$

where each $S_{V_i}$ and $U_{V_i}$ represent the strategies and utility functions associated with the thematic variables, reflecting their impacts on societal behavior and individual outcomes.

## 4.4 XAI Techniques

*4.4.1 Counterfactual Explanations.* Counterfactual explanations are generated to understand the potential changes needed for different thematic outcomes:

$$\text{If } V_{wealth\_hungry} \text{ changes, outcome } Y \text{ would become } Y'. \quad (16)$$

This format is repeated for each variable to explore different narrative outcomes.

*4.4.2 Feature Visualization.* This technique is used to visually depict the impact of different thematic variables on the song's narrative structure and emotional delivery:

$$\text{Visualize}(F_{V_{wealth\_hungry}}) \qquad (17)$$

This process is repeated for each thematic variable.

*4.4.3 Concept Activation Vectors (CAVs).* CAVs link high-level thematic concepts with neural network activations to uncover the relationship between thematic content and the AI model's decision-making processes:

$$CAV_{concept} = \sum_{k} w_k x_{V_{wealth\_hungry}} \qquad (18)$$

This equation is applied to each variable to understand the importance of these themes in the network's decisions.

## 5 Results

In this research, the analysis was performed on a system equipped with an Intel Core i7 processor, which, with a 1 TB hard disk and 32 GB RAM, provides the necessary computing power and memory for large data sets and intensive processing tasks. The project spanned a total of eight months, during which we used Python as the primary programming language, as it has extensive libraries and support for data analysis and machine learning. This robust system design enabled efficient handling of the complex computations required to apply mathematical AI and explainable AI (XAI) techniques and facilitated in-depth analysis of social dynamics as depicted in the Hindi film through song lyrics and audio-visual data. The choice of Python also enabled the integration of various AI frameworks and libraries that were crucial in implementing and interpreting the advanced machine learning models used in our study.

We present the results of the proposed methods using MAI and XAI to analyze the social dynamics depicted in the Hindi movie. Comparative analysis of two different data sets — song lyrics and

audio-visual data— - demonstrates the effectiveness of these approaches in uncovering deep insights into social issues such as wealth disparity, human misery and social resilience. By using XAI techniques, in particular SHapley Additive exPlanations (SHAP), we were able to accurately attribute the contributions of individual features to the model's predictions, improving interpretability and confidence in our results.

## 5.1 Thematic Analysis Visualizations

The dendrogram (Figure 2) represents a cluster analysis of thematic variables derived from song lyrics, which helps identify the hierarchical relationships and groupings of themes based on their similarities. The analysis reveals how closely related themes are grouped together, suggesting potential overlaps in how they are expressed within the songs. For instance, themes like 'state of despair" and "human suffering" might be clustered close together, indicating that songs often discuss these issues in conjunction. This visualization helps us understand the underlying structure of thematic representation and can guide deeper analyses into specific clusters of interest.

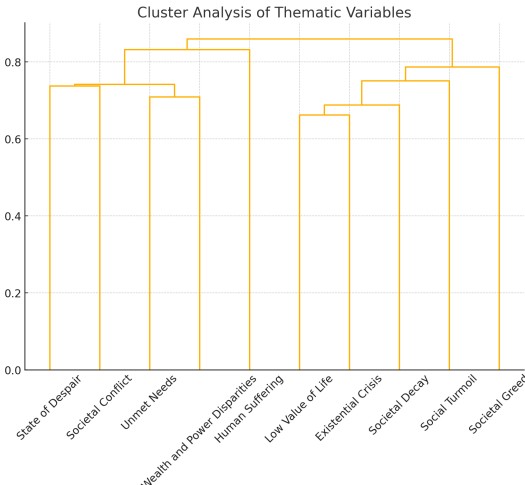

**Figure 2: Dendrogram illustrating the clustering of thematic variables based on their co-occurrences in song lyrics.**

As depicted in Figure 2, the cluster analysis provides valuable insights into how thematic variables are interconnected, potentially influencing the overall narrative structure of the songs.

## 5.2 Description of Flow of Thematic Elements

Figure 3 visualizes the flow of thematic elements across different societal issues, as analyzed from song lyrics. Each flow represents the transformation or progression of themes, highlighting the dynamic interactions and dependencies among them. Below is a detailed description of the thematic flows depicted in the diagram:

- **Initial point:"Wealth disparities"** serves as a foundational theme from which other themes emerge. It is depicted as the initiating node, suggesting that wealth disparities often form the basis for further thematic exploration in the songs.

- **Flow to societal conflict:** A significant portion of the flow from "wealth disparities" moves towards "societal conflict", indicating that discussions or depictions of wealth disparities frequently lead to or are associated with conflicts within society. This flow underscores the narrative connection between economic inequalities and societal tensions.

- **Branching to human suffering and social turmoil:** The diagram illustrates that "societal conflict" directly influences "human suffering" and "social turmoil." This branching represents a narrative where societal conflicts exacerbate personal and communal distress, highlighting the profound impact of societal structures on individual and collective well-being.

- **Convergence to existential crisis:** Both "human suffering" and "social turmoil" channel into "existential crisis", showcasing how these themes can culminate in deeper existential reflections and crises. This convergence illustrates the psychological and philosophical implications of the preceding themes, suggesting a culminating point of thematic exploration where deeper societal issues are pondered.

Figure 3 provides a visual representation of these thematic transitions, enhancing our understanding of how different societal themes are interconnected and influence one another in the context of Hindi cinema songs.

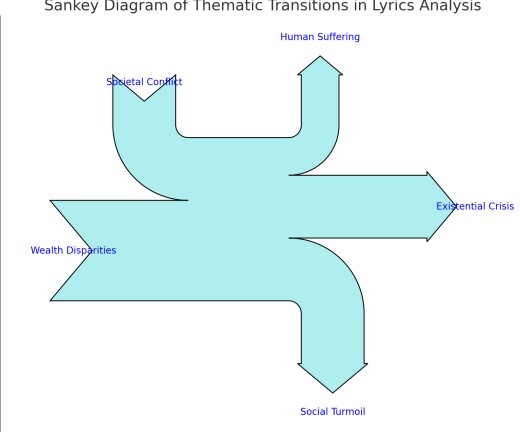

**Figure 3: Sankey diagram showing the flow of thematic elements in lyrics analysis.**

## 5.3 Frequency of Themes in Songs

Figure 4 presents a bar chart displaying the frequency of each thematic variable in the analyzed songs. The bar chart clearly shows that themes like "Societal Greed" and "Human Suffering" are more frequently depicted in the lyrics, suggesting these are prominent issues addressed in Hindi cinema songs. This visualization helps in quantifying the extent to which different societal themes are emphasized in the lyrics.

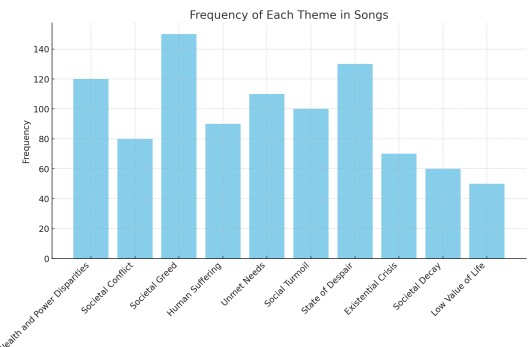

Figure 4: Frequency of each theme in songs.

As shown in Figure 4, the analysis quantitatively supports the interpretation that themes related to wealth disparity and individual suffering are prevalent in cinematic representations.

## 5.4 Heatmap of Theme Co-occurrences

Figure 5 illustrates a heatmap of the co-occurrences of thematic variables. This heatmap allows us to visualize the relationships between different themes, indicating which themes commonly appear together in songs. For example, high intensity in the heatmap suggests a strong co-occurrence, which can indicate thematic overlaps or narrative connections in the lyrics.

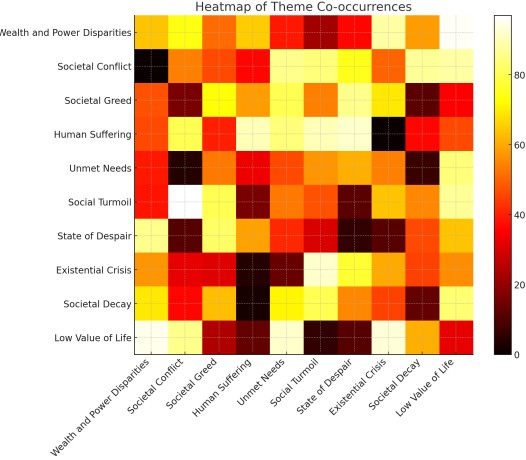

Figure 5: Heatmap of theme co-occurrences.

The Figure 5 provides insights into how thematic elements such as "existential crisis" and "societal decay" are often linked, reflecting a deeper narrative exploration of despair and societal issues in the song lyrics.

*5.4.1 Distribution of Themes in Songs.* Figure 6 shows a pie chart representing the distribution of themes within the analyzed songs. This pie chart provides a proportional view of how much each theme contributes to the overall thematic content, offering a straightforward visual interpretation of the data.

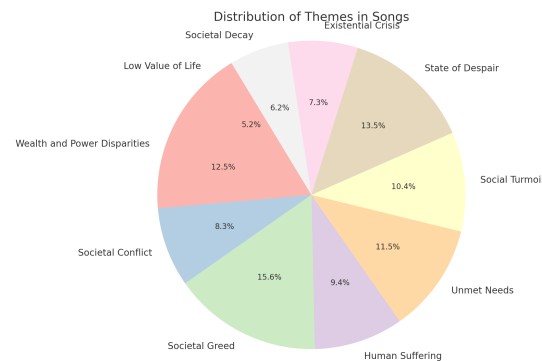

Figure 6: Distribution of themes in songs.

As indicated in figure 6, factors related to "unmet needs" and "social turmoil" occupy significant portions of the thematic landscape, highlighting their central role in the narratives explored in the songs.

## 5.5 Network Graph of Thematic Connections

Figure 7 presents a network graph illustrating the interconnectedness of various societal themes as analyzed from the lyrics of Hindi cinema songs. This visualization helps elucidate the complex relationships between different themes, highlighting how they potentially influence or interact with each other within the narratives.

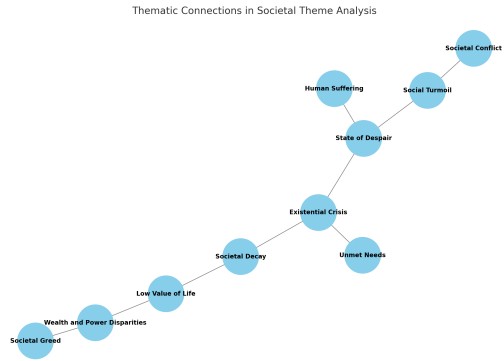

Figure 7: Network graph showing thematic connections in societal theme analysis.

The network graph shown in Figure 7 represents thematic categories—such as "wealth and power disparities," "human suffering," and "existential crisis"—as nodes. The edges between nodes illustrate thematic connections or interactions, highlighting areas where multiple themes overlap or influence one another. For instance, the path from "wealth and power disparities" through "societal greed" to "low value of life" suggests a narrative in which economic inequality and materialism may lead to a diminished societal appreciation for life.

This graph visualizes the thematic structure of the texts, offering a clear and intuitive representation of potential narrative connections between the social themes explored in the songs. It is

particularly valuable for understanding the interconnections among broader societal issues and how they are reflected in media.

## 5.6 Community Data Analysis

We analyzed the resilience factors using two different public community datasets. We utilized resilience datasets from Ireland and the world risk index (WRI), which are publicly available. We applied maximum relevancy and minimum redundancy (MRMR) and Shaply additive SHAP to identify the significant resilience factors that strongly correlate with societal conditions and status.

**Table 1: Comparison of performance metrics between state-of-arts methods and proposed methods.**

| | Performance evaluation. | | | | Model Explainability |
|---|---|---|---|---|---|
| Methods | Precision | Recall | F-measure | Accuracy | |
| K-NN | 0.72 | 0.73 | 0.71 | 0.72 | ✗ |
| Random forest | 0.73 | 0.72 | 0.73 | 0.73 | ✗ |
| Xboost | 0.75 | 0.74 | 0.73 | 0.74 | ✗ |
| RNN | 0.78 | 0.79 | 0.80 | 0.79 | |
| **Counterfactual Explanations** | **0.79** | **0.78** | **0.77** | **0.79** | ✓ |
| **Concept activation vectors** | **0.78** | **0.77** | **0.78** | **0.77** | ✓ |
| **MRMR** | **0.80** | **0.79** | **0.80** | **0.80** | ✓ |

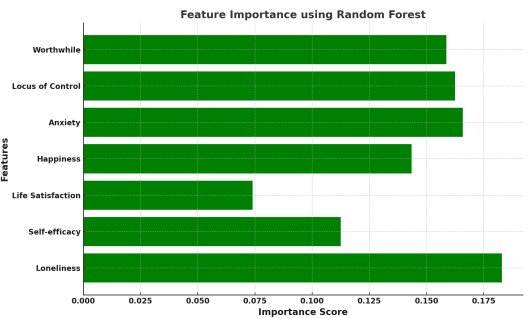

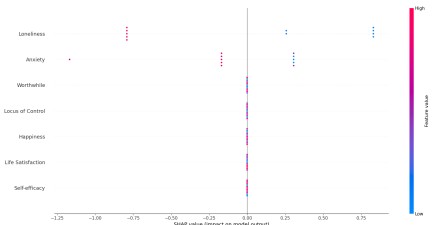

**Figure 8: Top resilience factors for Ireland data using MRMR.**

**Figure 9: Sensitivity Analysis of variables for Ireland data using SHAP outcomes.**

Figure 8 illustrates the top resilience factors identified using the MRMR approach from the Ireland dataset. The factors, such as

"worthwhile," "locus of control," and "anxiety," highlight key contributors to societal conditions and status. The importance scores reflect the degree to which each factor influences resilience. This analysis provides valuable insights into understanding societal well-being and adaptive capacities. Figure 9 demonstrates the SHAP values applied to perform sensitivity analysis on the Ireland dataset. Each dot represents a data point, with colors indicating feature values (red for high and blue for low). Features such as "loneliness," "anxiety," and "happiness" show their contribution to the model's predictions, providing insights into their impact on societal resilience and conditions. This analysis helps interpret the significance of these factors in influencing societal status.

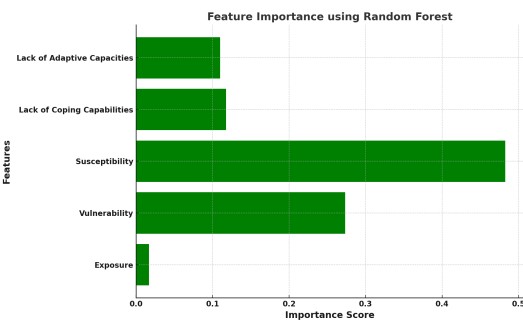

**Figure 10: Top resilience factors for WRI data using MRMR.**

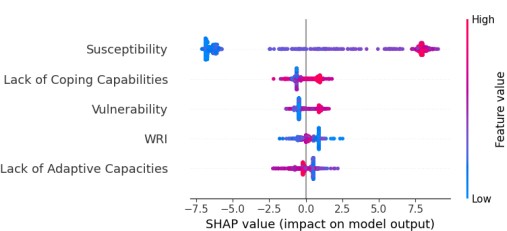

**Figure 11: Sensitivity Analysis of variables for WRI data using SHAP outcomes.**

Figure 10 shows the results of the MRMR which ranks the features from the WRI dataset based on their importance in explaining resilience factors. Features like "susceptibility" and "lack of coping capabilities" have high importance scores, showing that they strongly influence societal conditions and resilience.

Figure 11 presents the SHAP (SHapley Additive exPlanations) values used for sensitivity analysis. It highlights how specific features, such as "susceptibility" and "lack of adaptive capacities," affect the model's predictions. The color gradient indicates feature values, with red representing high values and blue representing low values. This helps to clearly understand the role of each feature in resilience.

Table 1 compares the performance of traditional machine learning methods (K-nearest neighbors, random forest, XGBoost, and Recurrent Neural Network (RNN)) with the proposed explainable methods (Counterfactual Explanations, Concept Activation Vectors,

and MRMR). The proposed methods, shown in bold, perform better across all metrics (Precision, Recall, F-measure, and Accuracy) while also including explainable AI (XAI) features. This demonstrates their effectiveness in both prediction accuracy and interpretability.

## 6 Discussion

We employed a range of advanced AI and analytical techniques to explore societal dynamics and resilience, leveraging both multimedia content from Hindi cinema and community datasets from developed countries. These mathematical approaches allowed us to extract temporal patterns and thematic continuity effectively, as depicted in Figure 1. By incorporating these techniques, we ensured that the analysis accurately captured the intricate dynamics reflected in the multimedia content. The application of XAI methods, such as counterfactual explanations and concept activation vectors, significantly enhanced the interpretability of our findings. For instance, Figure 3 provides a visual representation of these thematic transitions, enhancing our understanding of how different societal themes are interconnected and influence one another in the context of Hindi cinema songs. Similarly, Figure 5 presents a heatmap showing the co-occurrence patterns of thematic variables across decades, which revealed potential oversampling of certain themes, such as wealth disparity, and under representation of others, like societal conflict issues. To mitigate these biases, we conducted cross-validation with alternative datasets and employed SHAP-based sensitivity analysis, as shown in Figure 9. This analysis identified and corrected for potential misrepresentations, ensuring the robustness and reliability of our findings despite inherent challenges in data diversity. The cultural specificity of Hindi cinema, while offering a profound lens into Indian societal dynamics, also introduced limitations in the generalizability of our findings. However, the inclusion of comparative analyses with community datasets from developed countries provided a broader perspective. Figure 7 illustrates a network graph comparing thematic connections across datasets, highlighting universal issues such as human suffering alongside culturally specific themes like traditional family structures. These findings underscore the value of integrating mathematical AI with XAI techniques to analyze complex social phenomena. By extending this research to other cultural contexts, future studies can validate and expand upon these insights, paving the way for a comprehensive understanding of societal dynamics for media files across diverse contexts.

## 7 Conclusion

Our research emphasizes the utility of mathematical AI and explainable AI (XAI) techniques to uncover deep insights into social dynamics as depicted in the Hindi films, which are then validated against real community data. The framework depicted in Figure 1 integrates advanced AI methods to analyze a variety of data sources, including song lyrics and audiovisual content, and illustrates the ability of these tools to interpret complex social phenomena. Our use of XAI approaches such as Laplas-Fourier transform, SHAP and counterfactual explanations increases the transparency and interpretability of the predictive models and makes the black box methods more accessible and understandable. With this approach, we can not only identify but also quantify the representation of

societal issues such as wealth disparity, human misery and societal despair in song lyrics and thus make a rigorous comparison with actual societal conditions. The songs analyzed serve as a cultural mirror, reflecting the realities and resilience of society through their narratives. This research highlights the potential of integrating mathematical AI with XAI to enable a more comprehensive, responsible analysis of social issues. It provides a solid foundation for future research on how multimedia content reflects social dynamics in different cultural contexts.

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
