# OpenReview forum: "Mathematical AI-Driven Insights into Societal Dynamics and Resilience"
_ACM.org/TheWebConf/2025/Workshop/TIME — TIME 2025 Poster_

### Official Review · Reviewer_wmme · 2025-01-12
**Minor Revisions**

**Rating:** 6
**Confidence:** 4

**Review:**

Strengths -
1. good mathematical & empirical observations
2. data set details and evaluations

Weakness
1. Shallow literature review, use case study and future scope
2. More detailed and comprehensive statistical evaluations can further strengthen

---

### Official Review · Reviewer_PjZD · 2025-01-13

**Rating:** 4
**Confidence:** 1

**Review:**

I have to say, I think this paper cares about the issue to the extent that it should be published in NATURE, SCIENCE. I think this theme is too advanced and super crossover for me to review. But in my personal academic literacy, I think this paper is messed up.

---

### Official Review · Reviewer_xyBM · 2025-01-13
**Reviews of Mathematical AI-Driven Insights into Societal Dynamics and Resilience**

**Rating:** 5
**Confidence:** 4

**Review:**

This paper explores social dynamics and resilience by analyzing multimedia content from Hindi films and combining it with community data from developed countries, using Mathematical Artificial Intelligence (MAI) and Explainable Artificial Intelligence (XAI) techniques.

Advantages:

- The paper introduces a dataset consisting of 100 songs related to social issues from Hindi films since 1951, analyzing their audio, video, and lyrics, providing valuable research data.
- XAI techniques are employed to enhance the transparency and interpretability of the model.
- The findings offer valuable references for policymakers and practitioners.

Disadvantages:

- All formulas lack punctuation marks.
- The research findings may be limited by the cultural contexts of India and the selected developed countries, and their applicability to other cultural regions needs further validation and expansion. This limitation may not fully reflect the diversity of global social dynamics and resilience.
- Although many social theme variables were considered, there may still be other significant factors in real-world social systems that were not included in the study, such as unique customs in social cultures and the potential influence of religious beliefs on social behavior and resilience. These omissions could impact the comprehensiveness of the research findings.

---

### Official Review · Reviewer_xj6s · 2025-01-15
**The review evaluates a paper on the integration of AI techniques with multimedia content analysis to explore societal dynamics, focusing on Hindi cinema and community data, highlighting its quality, originality, and significance while noting areas for improvement.**

**Rating:** 7
**Confidence:** 3

**Review:**

Quality: The paper demonstrates a high level of quality in terms of its methodological approach and the integration of advanced AI techniques. The use of both traditional and cutting-edge AI methods, such as counterfactual explanations and concept activation vectors, indicates a robust analytical framework. The research is well-supported by data collected from Hindi cinema and community datasets, providing a comprehensive analysis of societal behaviors and resilience.

Clarity: The paper is clear in its objectives and methodology. It effectively communicates the integration of multimedia content analysis with AI techniques to explore societal dynamics. The use of figures and tables, such as SHAP values and performance metrics, aids in the clarity of the results and interpretations. However, some technical details, such as the specific implementation of AI models, could be elaborated for better understanding.

Originality: The work is original in its approach to combining multimedia content from Hindi cinema with community data to analyze societal behaviors. The novel use of AI techniques to interpret cultural artifacts and their impact on societal resilience is a significant contribution to the field. The interdisciplinary nature of the research, bridging cultural studies and AI, adds to its originality.

Significance: The research is significant as it provides new insights into the role of cultural artifacts in shaping societal behaviors and resilience. The findings have implications for academics, policymakers, and practitioners, offering a model for future research in global sociocultural studies. The ability to validate cinematic representations against real-world data enhances the study's relevance and applicability.

Pros:

Innovative integration of multimedia content analysis with AI techniques.
Comprehensive data collection from Hindi cinema and community datasets.
Clear presentation of results with visual aids like SHAP values and performance metrics.
Significant implications for understanding societal behaviors and resilience.
Interdisciplinary approach bridging cultural studies and AI.

Cons:

Some technical details regarding AI model implementation could be more detailed.
The paper may benefit from a more extensive discussion on the limitations and potential biases in data selection and analysis.
The focus on Hindi cinema may limit the generalizability of findings to other cultural contexts.

Overall, the paper presents a well-executed study with significant contributions to the understanding of societal dynamics through the lens of cultural artifacts and AI.

---

### Meta-Review · Area_Chair_42Kr · 2025-01-26

**Recommendation:** Accept (Poster)
**Confidence:** 3

**Metareview:**

This paper tackles an interesting and creative concept, using AI to analyze societal behaviors through the lens of Hindi cinema and real-world community data. It’s an innovative combination of cultural analysis AI techniques like SHAP, FOURIER Transforms, and counterfactual explanations. The framework presented is well-thought-out, and the analysis provides some fascinating insights into societal themes like wealth disparity and resilience.

While there’s room for improvement, like expanding the dataset beyond Hindi cinema and diving deeper into practical applications. The visuals, especially the thematic flows, and heatmaps are engaging and help articulate the findings well. It effectively demonstrates how AI and explainable techniques can uncover patterns in cultural and societal dynamics.

Overall - The paper’s strengths outweigh its weaknesses and brings a fresh take to the interdisciplinary research. It's worth accepting for its creativity and potential to inspire further work in this space.

---

### Decision · Program_Chairs · 2025-01-26

**Decision:**

Accept (Poster)

**Comment:**

The program chair concurs with the area chair's decision.

For the camera-ready version, please revise your paper according to the feedback provided by the reviewers.

Workshop papers must be written in English, follow a double-column format, and comply with the [ACM template](https://www2025.thewebconf.org/short-papers) and formatting guidelines. The template is also available in [Overleaf](https://www.overleaf.com/latex/templates/association-for-computing-machinery-acm-sig-proceedings-template/bmvfhcdnxfty). For authors using Microsoft Word, the Word Interim Template is recommended.

Camera-ready versions of accepted papers can and should include all information to identify authors, and should acknowledge any funding received that directly supported the presented research.

In addition, ensure that the DOI (to be provided by the PCs at a later stage) is included, and cite the workshop (to appear) using the following reference:

```
@inproceedings{time2025,
  title={TIME 2025: 1st International Workshop on Transformative Insights in Multi-faceted Evaluation},
  author={Lei Wang and Md Zakir Hossain and Syed Islam and Tom Gedeon and Sharifa Alghowinem and Isabella Yu and Serena Bono and Xuanying Zhu and Gennie Nguyen and Nur Haldar and Seyed Jalali and Abdur Razzaque and Imran Razzak and Rafiqul Islam and Shahadat Uddin and Naeem Janjua and Aneesh Krishna and Manzur Ashraf},
  booktitle={ACM Web Conference Workshop},
  year={2025}
}
```

Please note that at least one in-person registration is required for each accepted workshop paper to be included in the Companion Proceedings of WWW 2025. All accepted papers must be presented at the conference. Papers not presented (no-shows) may be withdrawn from the companion proceedings. Presentations will be conducted in two formats: oral and poster.

The camera-ready deadline for workshop papers is 7 February 2025 (AoE).